# PROMOBAL: PROTOTYPE-GUIDED MODALITY BALANCING IN MULTIMODAL CONTRASTIVE LEARNING

## ABSTRACT

Multimodal learning often suffers from *modality imbalance*, where dominant modalities overshadow weaker ones and unimodal encoders lack a shared representational goal. We propose a new end-to-end multimodal supervised contrastive learning framework, **Pro**totype-guided **Mo**dality contribution **Bal**ancing (**ProMoBal**), that integrates prototype-centered multimodal representation learning with sample-adaptive fusion. At its core, ProMoBal promotes a new *regular simplex geometry* for multimodal representation learning, where class prototypes are symmetrically arranged on a shared hypersphere that consistently spans both unimodal and fused representation spaces. This geometry provides a common reference for aligning unimodal and fused embeddings, while the proposed adaptive fusion mechanism mitigates modality balance on a per-sample basis. Extensive experiments with five benchmark datasets—spanning audio–video, image–text, and three-modality gesture recognition—show that ProMoBal consistently outperforms state-of-the-art multimodal supervised learning methods, achieving accuracy gains of up to 21% over unimodal baselines.

## 1 INTRODUCTION

Multimodal learning integrates heterogeneous data from different modalities to enhance a model's ability to perceive the world (Baltrušaitis et al., 2018). By jointly analyzing these diverse data types, multimodal models can extract complementary information and thus overcome the limitations of relying on a single modality (Joze et al., 2020). This paradigm has been widely applied to a wide range of tasks, such as action recognition (Shahroudy et al., 2017), emotion recognition (Song et al., 2022; Ranganathan et al., 2016), and audio-visual speech recognition (Mroueh et al., 2015; Oneață & Cucu, 2022). Yet, multimodal learning faces a key challenge, modality imbalance, where heterogeneous data often contain both dominant and weaker modalities (Huang et al., 2022; Fan et al., 2023). In the standard setup, features from different modalities are fused into a joint representation and supervision is provided only via the fused output. While simple and effective, this approach optimizes all encoders solely via a fusion loss, often allowing a dominant modality to drive learning and leaving weaker modalities undertrained, thereby exacerbating modality imbalance.

We highlight two key aspects of modality imbalance. *(1)* **Objective mismatch.** To mitigate modality imbalance, multitask-like approaches (Fan et al., 2023; Du et al., 2023; Wang et al., 2020) introduce unimodal losses alongside the fusion loss, thereby providing direct supervision to each encoder. This encourages each modality to learn discriminative features and promotes more balanced training across modalities. However, Wei & Hu (2024) argue that jointly optimizing unimodal and fusion objectives is inherently difficult, as a unimodal encoder cannot generally satisfy both, leading to conflicts during optimization. To address this, gradient modulation has been proposed to align the descent directions of unimodal and fusion losses (Wei & Hu, 2024), and a two-stage strategy has been introduced to separately optimize unimodal and fusion embeddings (Fan et al., 2024). Although partly effective, these approaches do not resolve the root cause—that a unimodal encoder cannot simultaneously satisfy both objectives. We conjecture that this limitation is partly because unimodal and fusion features pursue incoherent representational aims—unimodal features emphasize modality-specific discrimination, while fusion features emphasize modality-invariant representations—thereby inducing conflicts in their learning dynamics. Instead, if unimodal encoders are encouraged to capture modality-specific information, while both unimodal and fusion embeddings are simultaneously optimized toward a common goal, the embeddings may be jointly

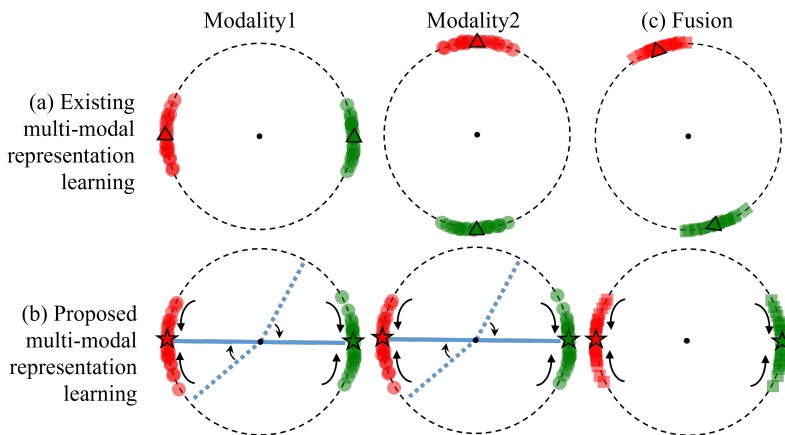

Figure 1: **Overview of proposed representation learning** (with two classes indicated by different colors; Modality 2 and 1 denote the dominant and weaker modalities, respectively)**. (a) A geometric configuration promoted by existing multimodal representation learning methods,** that may result in fused representations biased toward the dominant modality, Modality 2. Distortions in the dominant modality's geometry, if present, may propagate to fused features and degrade performance. (b) **The new configuration promoted by proposed ProMoBal** that jointly aligns unimodal (●) and *sample-adaptively* fused features (■) with shared prototypes (★), yielding fused representations that capture balanced modality-specific cues and modality-invariant structure. In addition, we align unimodal classifiers (skyblue sticks) to the prototypes. The configuration prevents fused features from being biased toward the dominant modality and mitigates distortions inherited from it.

learned without conflict, ultimately retaining modality-specific cues while also acquiring modality-invariant structure. *(2)* **Sample-specific modality contribution**: In realistic scenarios, the amount of effective information provided by each modality varies across samples (Wei et al., 2024). Conventional fusion strategies, e.g., concatenation, sum, and affine transformation, however, typically assign uniform weights across all samples, failing to capture these variations. For example, in an audio–video setting, if the majority of samples carry stronger cues in the audio modality, simple concatenation tends to bias the fusion toward audio. As a result, samples that rely more on video information become underrepresented, leading to suboptimal fusion embeddings. This underscores the importance of considering per-sample modality contributions in deriving fusion embeddings.

To overcome the two key challenges discussed above, we propose a novel end-to-end (E2E) multimodal supervised contrastive learning (MSCL) framework, **Pro**totype-guided **Mo**dality **Bal**ancing, referred to as **ProMoBal**. Our aim is to promote the new *prototype-centered geometric configuration* in Figure 1(b) in which both unimodal and fused representations are jointly aligned. To this end, we propose the following two complementary strategies:

- **Unified prototype-based objective.** We align both unimodal and fusion embeddings with *shared prototypes*—our *common goal*—while preserving modality-specific information in unimodal encoders. This reduces objective conflicts and ultimately promotes representations that retain modality-specific cues while capturing modality-invariant structure.
- **Sample-adaptive fusion.** We introduce a *sample-adaptive balancing mechanism* that adjusts modality contributions at both the sample level and the feature-dimension level, thereby mitigating imbalance between dominant and weaker modalities.

These enable **ProMoBal** to produce fused representations that integrate complementary cues from modalities in a balanced manner while maintaining invariance to modality differences, particularly on a per-sample basis. Our extensive experiments with five benchmark datasets demonstrate that

- The **ProMoBal** framework consistently outperforms existing state-of-the-art (SOTA) multimodal supervised learning baselines. In particular, ProMoBal achieves consistent improvements even on a dataset with three modalities and across diverse modality types, highlighting its broad applicability.

Further empirical analysis demonstrates that ProMoBal promotes modality-balanced and class-balanced representations, and smoother decision boundaries.

## 2 RELATED WORKS

**Modality imbalance in multimodal learning.** First, we review existing multimodal supervised learning methods specifically designed to address the modality-imbalance problem. Among them, OGM-GE (Peng et al., 2022), AGM (Li et al., 2023), MMPareto (Wei & Hu, 2024), and LFM (Yang et al., 2024) adopt gradient modulation; PMR (Fan et al., 2023) employs prototype alignment regularization; ReconBoost (Hua et al., 2024) and MLA (Zhang et al., 2024) alternate modality updates during training; and DI-MML (Fan et al., 2024) adopts a two-stage training strategy, separately optimizing unimodal and fusion embeddings. The above existing methods *overlook* two important factors: *(1)* the objective mismatch between unimodal and fusion embeddings, and *(2)* the sample-specific variability in modality contributions. The proposed ProMoBal framework aims to mitigate these issues by jointly optimizing unimodal and fused embeddings under a common prototype-centered representational goal, while balancing modality contributions on a per-sample basis.

**Multimodal contrastive learning (CL).** To pretrain multimodal models (specifically, unimodal encoders and a fusion module), multimodal self-supervised CL has been widely adopted (Radford et al., 2021), where paired multimodal samples are treated as positive while others are treated as negatives, i.e., an instance-level CL. In the supervised setting, methods such as GMC (Poklukar et al., 2022), DI-MML (Fan et al., 2024), and LFM (Yang et al., 2024) also employ instance-level CL. Unlike self-supervised approaches, they additionally use class labels to jointly train either unimodal classifiers or a fusion classifier. Although these approaches use CL to exploit cross-modal interactions, they do *not* consider the per-sample modality contributions in fusing unimodal embeddings. As a result, the fusion embeddings tend to follow the dominant modality, as illustrated in Figure 1(a). Different from the MSCL methods above, the proposed ProMoBal framework employs a *class-level* CL approach: within each modality, samples from the same class are treated as positives and those from different classes as negatives, with the contrastive objective applied independently per modality (i.e., no cross-modal pairing). We further augment this objective with shared class prototypes that act as anchors, providing a common reference for representation learning. Moreover, by additionally regulating modality contributions via sample-level fusion embedding generation, we encourage the representations to converge toward the new geometric configuration in Figure 1(b).

## 3 METHODS

We outline the problem formulation and overall architecture of ProMoBal in §3.1, and present its key innovations in unimodal representation learning, fused representation learning, and classifier alignment in §3.2, §3.3, and §3.4, respectively.

### 3.1 OVERVIEW

#### 3.1.1 PROBLEM FORMULATION

Our general goal is to learn a mapping from the input space $\mathcal{X}$ to the target space $\mathcal{Y} = \{1, 2, .., C\}$, where $C$ is the number of classes. Each sample $\mathbf{x}^i$ consists of $M$ modality-specific components: $\mathbf{x}^i = \{\mathbf{x}^i_m\}_{m=1}^M$, where $\mathbf{x}^i_m$ denotes the $m$th modality of the $i$th input. Each modality has a corresponding encoder $f_m(\cdot)$ that extracts a feature representation $\mathbf{z}^i_m \in \mathbb{R}^D$ from the input $\mathbf{x}^i_m$, where $D$ is the feature dimension. Each modality is further equipped with a linear classifier $g_m(\cdot)$ that outputs a logit vector $\mathbf{l}^i_m \in \mathbb{R}^C$ from $\mathbf{z}^i_m$, i.e., $\mathbf{l}^i_m = g_m(\mathbf{z}^i_m) = \mathbf{W}_m \mathbf{z}^i_m + \mathbf{b}_m$. Here, $\mathbf{W}_m = [(\mathbf{w}^1_m)^\top, \ldots, (\mathbf{w}^C_m)^\top]^\top \in \mathbb{R}^{C \times D}$ denotes the classifier weight matrix for the $m$th modality, and $\mathbf{b}_m \in \mathbb{R}^C$ is the corresponding bias vector. We derive a fusion feature $\mathbf{z}^i_{\text{fus}} \in \mathbb{R}^D$ from unimodal features: $\mathbf{z}^i_{\text{fus}} = \Phi(\mathbf{z}^i_1, \ldots, \mathbf{z}^i_M)$, where we design a sample-adaptive fusion module $\Phi(\cdot)$. In CL, we $\ell_2$-normalize all representations to unit length. We produce logits from the fusion feature by computing its similarity with shared class prototypes $\mathbf{P} = [\mathbf{p}_1^\top, \ldots, \mathbf{p}_C^\top]^\top \in \mathbb{R}^{C \times D}$: $\mathbf{l}^i_{\text{fus}} = \mathbf{P} \mathbf{z}^i_{\text{fus}} \in \mathbb{R}^C$. We learn ProMoBal in a supervised learning manner with $\{(\mathbf{x}^i, y_i) : \forall i\}$, where we define our common goal in learning unimodal and fused embeddings as follows:

**Definition** (Common goal). *In the context of learning unimodal and fused embeddings $\{\mathbf{z}^i_m, \mathbf{z}^i_{\text{fus}} : m = 1, \ldots, M, \forall i\}$, we define the* common goal *as the alignment of both feature types with the same set of shared class prototypes $\{\mathbf{p}_c : c = 1, \ldots, C\}$.*

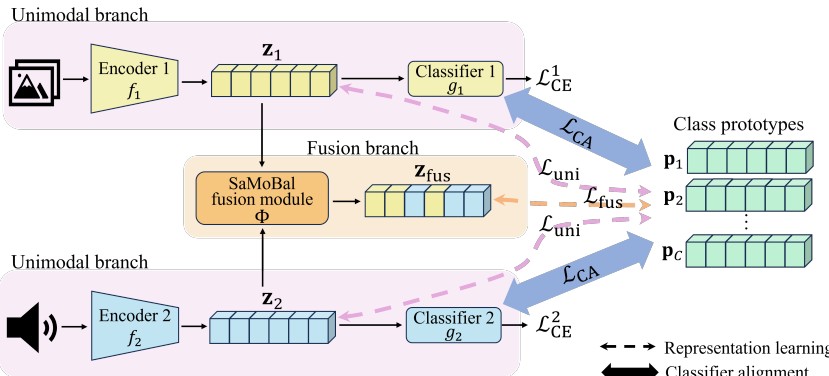

Figure 2: **Overall architecture of ProMoBal** (proposed MSCL with $M = 2$). We overview all proposed innovations—unimodal and fused representation learning losses $\{\mathcal{L}_{\text{uni}}, \mathcal{L}_{\text{fus}}\}$, per-sample fusion module $\Phi$, and classifier alignment loss $\mathcal{L}_{\text{CA}}$—in §3.1.2. All representations $\{\mathbf{z}_1, \mathbf{z}_2, \mathbf{z}_{\text{fus}}\}$ are $\ell_2$-normalized in CL.

The goal of the proposed ProMoBal framework is to jointly optimize unimodal and fusion features with respect to the common goal defined above, while forming fusion features by dynamically balancing modality contributions on a per-sample basis. Figure 1(b) illustrates our geometric objective in representation learning.

### 3.1.2 OVERALL ARCHITECTURE

We design an E2E architecture to achieve the above goal, as illustrated in Figure 2. We propose the following key components in ProMoBal:

*(1)* **Unimodal representation learning:** We propose a **Prototype-based Unimodal Contrastive Learning (Pro-UCL)** loss $\mathcal{L}_{\text{uni}}$ that encourages unimodal embeddings to cluster around shared prototypes and drives the embeddings of each class to collapse to the vertices of a shared regular simplex[1]. This preserves modality-specific discrimination via supervised CL within each modality while aligning all unimodal embeddings under the common goal. In addition, we equip each unimodal encoder with an individual cross-entropy (CE) objective $\mathcal{L}_{\text{CE},m}$. See the unimodal branches with $\{\mathcal{L}_{\text{uni}}, \mathcal{L}_{\text{CE},m} : m = 1, 2\}$ in Figure 2.

*(2)* **Fused representation learning:** We propose a **Sample-Adaptive Modality Contribution Balancing (SaMoBal)** fusion module $\Phi$ that adjusts *per-sample*, *per-dimension* contributions of each modality in deriving fused embeddings, by promoting their alignment with the prototype of the target class. This adaptive mechanism mitigates modality imbalance by preventing dominant modalities from overwhelming weaker ones, yielding more balanced fusion embeddings. In addition, we propose the **Prototype–Fusion Distillation (Pro-FD)** loss that distills a prototype-based reference distribution encoding the relational structure among prototypes into fused embeddings. This further encourages the fused representations to capture modality-invariant information encoded in shared prototypes. See the fusion branch with $\mathcal{L}_{\text{fus}}$ in Figure 2.

*(3)* **Classifier alignments:** We propose a **Classifier Alignment (CA)** loss $\mathcal{L}_{\text{CA}}$ that aligns unimodal classifier weights with shared prototypes, a unified reference for the common goal. See the classifier alignments in Figure 2.

### 3.2 UNIMODAL REPRESENTATION LEARNING

To learn modality-specific discrimination rather than being dominated by a fusion loss, we equip each unimodal encoder with an individual classifier and CE loss $\mathcal{L}_{\text{CE},m}$, $m = 1, \dots, M$ (Fan et al., 2023; Du et al., 2023; Wang et al., 2020; Fan et al., 2024). However, if only this approach is used, geometric configurations of unimodal representations across different modalities are misaligned; see Figure 1(a). To achieve an aligned configuration across modalities as illustrated in Figure 1(b), we introduce a new unimodal supervised CL approach in the following section.

---

[1]In the proposed regular simplex configuration for multimodal learning, shared prototypes of different classes are placed on the hypersphere in a symmetric and equidistant manner. See Figure 1(b).

### 3.2.1 PRO-UCL: PROTOTYPE-BASED UNIMODAL CONTRASTIVE LEARNING

We propose the Pro-UCL loss $\mathcal{L}_{\text{uni}}$, a prototype-augment variant of supervised contrastive learning (Khosla et al., 2020) for multimodal learning, where we explicitly incorporate shared class prototypes along with multiple positive samples to guide unimodal representations:

$$\mathcal{L}_{\text{uni},m} := -\frac{1}{|\mathcal{B}|} \sum_{i \in \mathcal{B}} \frac{1}{|\mathcal{P}(i)|} \sum_{j \in \mathcal{P}(i)} \log \frac{\exp(\mathbf{z}_m^i \cdot \mathbf{z}_m^j / \tau)}{\sum_{c=1}^{C} \frac{1}{|\mathcal{B}(c)|} \sum_{k \in \mathcal{B}(c)} \exp(\mathbf{z}_m^i \cdot \mathbf{z}_m^k / \tau)}, \quad \forall m, \quad (1)$$

and $\mathcal{L}_{\text{uni}} = \frac{1}{M} \sum_{m=1}^{M} \mathcal{L}_{\text{uni},m}$, where $i$ denotes the index of an anchor sample, $\mathcal{B} = \{1, \ldots, N\}$ denote the index set of all samples in a batch, and the notation $|\cdot|$ indicates the cardinality of a set. Here, we define the set of indices of all positive samples in a batch, including the index of the corresponding prototype but excluding the anchor index $i$: $\mathcal{P}(i) := \{n \in \widetilde{\mathcal{B}} \setminus \{i\} : y_n = y_i\}$, where $\widetilde{\mathcal{B}} := \mathcal{B} \cup \{N+1, \ldots, N+C\}$ denotes augmented $\mathcal{B}$ with shared class prototypes $\{\mathbf{p}_1, \ldots, \mathbf{p}_C\}$. We define the set of indices of all samples with class $c$ in a batch, including the corresponding prototype index: $\mathcal{B}(c) := \{n \in \widetilde{\mathcal{B}} : y_n = c\}, c \in \mathcal{Y}$. $\tau \in \mathbb{R}_{>0}$ is temperature parameter. By including shared prototypes in every batch, we can effectively handle all classes, even if some are absent from a batch.

Extending results in (Graf et al., 2021; Zhu et al., 2022), we obseve the following for (1) ($\tau = 1$):

$$\mathcal{L}_{\text{uni}} \geq \frac{1}{M|\mathcal{B}|} \sum_{m=1}^{M} \sum_{i \in \mathcal{B}} \log \left(1 + (C-1) \exp \left( \underbrace{\frac{1}{C-1} \sum_{c \in \mathcal{Y} \setminus \{y_i\}} \frac{1}{|\mathcal{B}(c)|} \sum_{j \in \mathcal{B}(c)} \mathbf{z}_m^i \cdot \mathbf{z}_m^j}_{\text{repulsion incl. shared prototypes}} - \underbrace{\frac{1}{|\mathcal{P}(i)|} \sum_{k \in \mathcal{P}(i)} \mathbf{z}_m^i \cdot \mathbf{z}_m^k}_{\text{attraction incl. shared prototype}} \right) \right),$$

where $\mathcal{Y} \setminus \{y_i\}$ indicates the set of all classes except $y_i$ ($i$th anchor's class). This observation suggests that each unimodal representation is attracted to the corresponding shared class prototype and repelled from the others. Minimizing the proposed Pro-UCL loss in (1) can promote that unimodal embeddings of each class collapse to their shared class prototypes and eventually the vertices of a regular simplex (Papyan et al., 2020; Zhu et al., 2022). This symmetric geometry configuration has been empirically shown to provide important benefits, such as improved generalization performance (Papyan et al., 2020), by providing a balanced feature space across classes.

As shown above, the $\mathcal{L}_{\text{uni},m}$ loss in (1) uses positive and negative samples within each modality to promote attraction and repulsion, respectively, thereby enabling each unimodal encoder to learn *modality-specific* discriminative representations. Moreover, by incorporating shared prototypes, the $\mathcal{L}_{\text{uni}}$ loss in (1) further aligns unimodal embeddings across modalities under a common reference, encouraging a coherent geometry that facilitates *modality-invariant* representation learning.

**Class averaging.** The proposed loss in (1) uses the *class averaging* mechanism (Zhu et al., 2022) in its denominator to improve the effectiveness of incorporating shared prototypes in each class. Without this mechanism—i.e., when naively incorporating shared prototypes into the standard supervised contrastive learning framework (Khosla et al., 2020)—each class typically contains many samples but only a single prototype, causing a repulsion term to be dominated by sample contributions.

## 3.3 FUSED REPRESENTATION LEARNING

We propose SaMoBal module $\Phi$ that derives fused embeddings by adjusting contributions of each modality on both *per-sample* and *per-dimension* basis, along with its training & inference processes.

### 3.3.1 SAMOBAL: SAMPLE-ADAPTIVE MODALITY CONTRIBUTION BALANCING (TRAIN)

To learn balanced fused representations, our key idea is to derive fusion embeddings by adaptively selecting, for each sample and dimension, the unimodal embeddings that best align with a prototype from the ground-truth (GT) class, referred to as the GT prototype. See Figure 3(a). Specifically, for each training sample $i$ and each dimension $d$, we compare unimodal features $\{(\mathbf{z}_m^i)_d : m = 1, \ldots, M\}$ with the corresponding GT class prototype $(\mathbf{p}_{c=y_i})_d$, and select the most similar features among modalities to construct a fused embedding $(\mathbf{z}_{\text{fus}}^i)_d$, for $d = 1, \ldots, D$.

To make this hard selection process differentiable, we use the Gumbel-Softmax (Jang et al., 2016) that approximates the non-differentiable $\mathrm{argmax}$ operation during training by injecting Gumbel noise and using a softmax with temperature, enabling gradients to flow, while still yielding hard selections in the forward pass. In particular, we use the straight-through estimation trick (Van Den Oord et al., 2017). In forward pass, for each sample $i$ and dimension $d$, we apply Gumbel-Softmax to the similarity scores $\{s_{m,d}^i = \mathrm{Sim}((\mathbf{z}_m^i)_d, (\mathbf{p}_{y_i})_d) : \forall m\}$

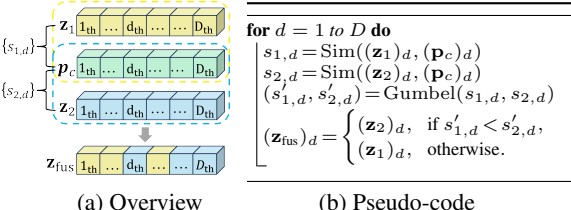

(a) Overview      (b) Pseudo-code

Figure 3: **An illustration and pseudo-code for the SaMoBal module in fusion learning** (for each sample; $M=2$, $c=y_i$)**.** The module adaptively adjusts the modality contribution at the feature-dimension level. Here, $\mathrm{Gumbel}(\cdot)$ denote the Gumbel-softmax.

to produce a near one-hot vector of size $M$, and then apply $\mathrm{argmax}$ to select the modality feature with the highest score, where $\mathrm{Sim}(\cdot)$ denotes a similarity function (here, instantiated as cosine similarity in our implementation). The pseudo-code in Figure 3(b) illustrates this process (sample index $i$ omitted for clarity). In the backward pass, gradients are computed through the soft Gumbel-Softmax distribution, allowing the model to approximate hard selections while remaining fully differentiable. With this scheme, we preserve discrete choices while still allowing gradients to flow through all modality features, including those not selected, thereby enabling effective E2E training.

We aim to optimize the SaMoBal module so that it learns to select modality features most consistent with the correct class. Yet, alignment to the GT prototype alone does not ensure that fused embeddings respect the relative geometry among all prototypes. The next section addresses this.

### 3.3.2 PRO-FD: PROTOTYPE–FUSION DISTILLATION

We propose the Pro-FD loss $\mathcal{L}_{\mathrm{fus}}$ that distills the prototype-based reference distribution into the fused embedding. Unlike hard CE supervision, this reference distribution encodes the *geometric relational structure among prototypes*, enabling the fused representation to learn not only the target class but also its relative position within the prototype space.

We first define the following two distributions for each sample $i$:

$$\psi_{\mathrm{ref}}^i(c) := \frac{\exp(\mathbf{p}_{y_i} \cdot \mathbf{p}_c)}{\sum_{c'=1}^{C} \exp(\mathbf{p}_{y_i} \cdot \mathbf{p}_{c'})} \quad \text{and} \quad \psi_{\mathrm{fus}}^i(c) := \frac{\exp(\mathbf{z}_{\mathrm{fus}}^i \cdot \mathbf{p}_c)}{\sum_{c'=1}^{C} \exp(\mathbf{z}_{\mathrm{fus}}^i \cdot \mathbf{p}_{c'})}, \tag{2}$$

for $c = 1, \ldots, C$. Here, *(1)* $\{\psi_{\mathrm{ref}}^i(c) : \forall c\}$ denotes the **reference distribution**, which for each sample $i$ encodes the geometric relational structure among prototypes $\{\mathbf{p}_c : \forall c\}$ by evaluating their relative similarities to the ground-truth prototype $\mathbf{p}_{y_i}$; and *(2)* $\{\psi_{\mathrm{fus}}^i(c) : \forall c\}$ denotes the **fused embedding distribution**, which for each sample $i$ represents the class likelihoods predicted from the fused embedding $\mathbf{z}_{\mathrm{fus}}^i$ with respect to the prototype space.

To align the fused distribution $\psi_{\mathrm{fus}}^i$ with the reference $\psi_{\mathrm{ref}}^i$, we adopt Sinkhorn Knowledge Distillation (SinKD) instead of the conventional Kullback-Leibler (KL) divergence. Unlike KL that ignores geometric relationships among prototypes and may mislead optimization when prototypes are correlated, SinKD formulates alignment as an optimal transport problem based on *pairwise* prototype distances. This enables $\psi_{\mathrm{fus}}^i$ to capture both the target prototype and its relative position to others, thus reflecting the global prototype geometry encoded in $\psi_{\mathrm{ref}}^i$.

Finally, we define the Pro-FD loss by $\mathcal{L}_{\mathrm{fus}} := \frac{1}{|\mathcal{B}|} \sum_{i=1}^{|\mathcal{B}|} \mathrm{SinKD}(\{\psi_{\mathrm{fus}}^i(c)\}_{c=1}^{C}, \{\psi_{\mathrm{ref}}^i(c)\}_{c=1}^{C})$. This loss encourages the fused representations to form class-balanced prototype-centered structures, where no single class prototype dominates the alignment and inter-class relations are proportionally preserved. By reflecting the global prototype geometry, the fused embeddings develop smoother decision boundaries and demonstrate stronger generalization in multimodal fusion.

### 3.3.3 SAMOBAL (INFERENCE)

In training, SaMoBal aligns fused embeddings with the GT prototype, thereby learning a selection rule that favors the correct class. At inference, however, the GT label is unknown. We apply the same rule to every prototype, producing a set of candidate fused embeddings. Among these, we choose

the candidate that shows the strongest alignment with the prototype from which it was derived, as the final fused representation. This procedure mirrors the selection behavior obtained from training.

Specifically, we first apply the same selection mechanism with every prototype $\mathbf{p}_c$, yielding $C$ candidate fusion embeddings for each sample $i$, $\{\mathbf{z}_{\text{fus},c}^i : c = 1, \ldots, C\}$.[2] Next, we score each candidate $\mathbf{z}_{\text{fus},c}^i$ by measuring its strongest alignment with the prototype bank $\mathbf{P}$, and we choose the candidate with the highest score as the final fused embedding:

$$\hat{\mathbf{z}}_{\text{fus}}^i = \mathbf{z}_{\text{fus},\hat{c}}^i, \quad \hat{c} = \operatorname*{argmax}_{c=1,\ldots,C} r_c, \quad \{r_c = \max_{k=1,\ldots,C} \mathbf{p}_k^\top \mathbf{z}_{\text{fus},c}^i : c = 1, \ldots, C\}. \tag{3}$$

Finally, we compute prediction logits from the selected fused embedding: $\hat{\mathbf{l}}_{\text{fus}}^i = \mathbf{P}\hat{\mathbf{z}}_{\text{fus}}^i$.

### 3.4 CA: Classifier Alignments

We re-frame an ideal geometric configuration in single-modal learning (Papyan et al., 2020) for multimodal learning. See the illustration in Figure 1(b). Via the designs proposed in §3.2–3.3, we can promote the new representational geometric configuration illustrated in Figure 1(b). Yet, the proposed designs may *not* sufficiently promote that unimodal classifiers are aligned with shared prototypes; see their geometric illustrations in Figure 1(b).

In single-modal learning, the alignment between class prototypes, e.g., class means, and weights of a linear classifier is a defining property of the ideal geometric configuration (Papyan et al., 2020). To promote an analogous alignment property in the multimodal setting, we propose the CA loss $\mathcal{L}_{\text{CA}}$ that aligns unimodal linear classifiers with shared prototypes:

$$\mathcal{L}_{\text{CA}}^m := \left\| \|\mathbf{W}_m\|_{\text{F}}^{-1}\mathbf{W}_m - \|\mathbf{P}\|_{\text{F}}^{-1}\mathbf{P} \right\|_{\text{F}}^2, \tag{4}$$

and $\mathcal{L}_{\text{CA}} = \frac{1}{M}\sum_{m=1}^M \mathcal{L}_{\text{CA}}^m$, where $\|\cdot\|_{\text{F}}$ denotes the Frobenius norm.

**Geometric properties:** In conjunction with the approaches in §3.2–3.3, this alignment induces a symmetric geometry across the representation and classifier spaces, in both unimodal and fused learning, thereby realizing the overall proposed configuration in Figure 1(b). This implies that unimodal and fused representations can take different forms to capture modality-specific and/or invariant information, yet their directions on the unit sphere are encouraged to align, and their classification outcomes are promoted to be consistent across unimodal and fusion settings.

### 3.5 Overall ProMoBal loss

Our overall loss is given by $\mathcal{L} := \lambda_{\text{uni}}\mathcal{L}_{\text{uni}} + \lambda_{\text{CE}}\mathcal{L}_{\text{CE}} + \lambda_{\text{fus}}\mathcal{L}_{\text{fus}} + \lambda_{\text{CA}}\mathcal{L}_{\text{CA}}$, where $\{\lambda_{\text{uni}}, \lambda_{\text{CE}}, \lambda_{\text{fus}}, \lambda_{\text{CA}}\}$ are hyperparameters that control the balance among losses. Here, $\mathcal{L}_{\text{CE}} := \frac{1}{M}\sum_{m=1}^M \mathcal{L}_{\text{CE}}^m$.

**Effects:** Our framework produces a balanced and geometry-aware multimodal representation with the following effects: *(1) modality balance* is achieved by the SaMoBal module, which adaptively regulates per-sample, per-dimension contributions of each modality; *(2) modality-specific discrimination* is preserved by unimodal CE losses and the Pro-UCL objective that promote class separation within each modality; *(3) Modality-invariant information* merges as unimodal embeddings are aligned to shared prototypes by Pro-UCL and fused embeddings are further distilled by Pro-FD to reflect the prototype geometry; *(4) class balance* arises from the prototype-centered geometry promoted by Pro-UCL and Pro-FD, preventing dominance by any single class and promoting uniform representation across all classes.

Together with the geometric properties in the previous section, these effects yield representations that are discriminative, invariant, balanced, and geometrically consistent across modalities.

## 4 Results and discussion

This section presents our main experimental results and discussions. In addition to the main results presented in the paper, we provide ablation studies on the four primary components of ProMoBal, an analysis of the SaMoBal module with different selection schemes, etc. in §B–E of the appendix.

---

[2]Following convention, we replace the Gumbel-Softmax used in training with a standard softmax during inference.

Table 1: Performance comparisons with different multimodal learning baselines.

| Methods | CREMA-D | | KineticsSounds | | Sarcasm | | Twitter2015 | | NVGesture | |
| | ACC | MAP | ACC | MAP | ACC | $F_1$ | ACC | $F_1$ | ACC | $F_1$ |
|---|---|---|---|---|---|---|---|---|---|---|
| Unimodal 1 | 63.17 | 68.61 | 54.12 | 56.69 | 81.36 | 80.65 | 73.67 | 68.49 | 78.22 | 78.33 |
| Unimodal 2 | 45.83 | 58.79 | 55.62 | 58.37 | 71.81 | 70.73 | 58.63 | 43.33 | 78.63 | 78.65 |
| Unimodal 3 | - | - | - | - | - | - | - | - | 81.54 | 81.83 |
| Concat. | 63.31 | 68.41 | 64.55 | 71.31 | 82.86 | 82.43 | 70.11 | 63.86 | 81.33 | 81.47 |
| Sum | 63.44 | 69.08 | 64.97 | 71.03 | 82.94 | 82.47 | 73.12 | 66.61 | 82.99 | 83.05 |
| GMC | 65.99 | 70.62 | 65.87 | 71.39 | 84.06 | 83.40 | 74.54 | 68.88 | 83.12 | 83.41 |
| OGM-GE | 66.94 | 71.73 | 66.06 | 71.44 | 83.23 | 82.66 | 74.92 | 68.74 | - | - |
| PMR | 66.59 | 70.36 | 66.56 | 71.93 | 83.61 | 82.49 | 74.25 | 68.62 | - | - |
| AGM | 67.07 | 73.58 | 66.02 | 72.52 | 84.28 | 83.44 | 74.83 | 69.11 | 82.78 | 82.82 |
| ReconBoost | 74.84 | 81.24 | 70.85 | 74.24 | 84.37 | 83.17 | 74.42 | 68.34 | 84.13 | 86.32 |
| MMPareto | 74.87 | 75.15 | 70.00 | 78.50 | 83.48 | 82.84 | 73.58 | 67.29 | 83.82 | 84.24 |
| MLA | 79.43 | 85.72 | 70.04 | 74.13 | 84.26 | 83.48 | 73.52 | 67.13 | 83.73 | 83.87 |
| LFM | 83.62 | 90.06 | 72.53 | 78.38 | 84.97 | 84.57 | 75.01 | 70.57 | 84.36 | 84.68 |
| DI-MML | 82.34 | 88.25 | 70.64 | 75.72 | 85.32 | 85.13 | 74.83 | 68.61 | - | - |
| **ProMoBal** | **85.68** | **91.85** | **76.6** | **81.22** | **89.17** | **88.65** | **77.08** | **72.52** | **86.84** | **86.97** |

## 4.1 EXPERIMENTAL SETUPS

We compared the proposed ProMoBal framework with numerous multimodal supervised learning methods, including three SOTA MSCL methods—DI-MML (Fan et al., 2024), LFM (Yang et al., 2024), and GMC (Poklukar et al., 2022), across five benchmark multimodal classification datasets.

**Datasets:** For audio–video multimodal classification, we used CREMA-D (Cao et al., 2014) and Kinetics-Sounds (Arandjelovic & Zisserman, 2017). For image–text multimodal classification, we used Sarcasm (Cai et al., 2019) Twitter2015 (Yu & Jiang, 2019). For reg-green-blue (RGB)–depth–optical flow (OF) multimodal analysis, we used NVGesture (Molchanov et al., 2016). See details of the datasets, implementation, and preprocessing in §A in the appendix.

**Baseline settings:** Unimodal 1 and Unimodal 2 correspond to audio and video in the audio–video datasets, and to image and text in the image–text datasets. For the NVGesture dataset, Unimodal 1, Unimodal 2, and Unimodal 3 denote RGB, OF, and depth modalities, respectively. In the Concat. and Sum settings, fusion embeddings were obtained by concatenation and summation, respectively, under the standard setup that uses only the fusion CE loss. Following Yang et al. (2024), we used the default unimodal encoders, based on convolutional neural networks or transformers, depending on the dataset—e.g., ResNet (He et al., 2016), I3D (Carreira & Zisserman, 2017), and BERT (Devlin et al., 2019). All existing multimodal learning methods are referenced in §2.

**Evaluation metrics:** Following Yang et al. (2024), we used accuracy (ACC) and mean average precision (MAP) for the audio-video datasets, and ACC and Macro F1 score ($F_1$) for the image-text and RGB-depth-OF datasets. ACC measures the proportion of correctly classified samples. MAP is the mean of the average precision scores computed across all classes. The $F_1$ score is the average of the class-wise F1 scores, reflecting the model's overall balance between precision and recall. All metrics are reported as percentages (%).

## 4.2 COMPARISONS BETWEEN DIFFERENT MULTIMODAL SUPERVISED LEARNING METHODS

Experimental results in Table 1[3] demonstrate that the proposed ProMoBal framework consistently outperforms all SOTA multimodal learning baselines across datasets, regardless of the unimodal encoder architectures. Specifically, the proposed ProMoBal improves accuracy by more than 2% and up to 4% across datasets, compared with existing multimodal learning baselines. Even in the three-modality setup (i.e., NVGesture), ProMoBal outperformed existing multimodal learning baselines, highlighting the broad applicability of the proposed method. On the Twitter2015 dataset—where existing multimodal learning baselines show little or no gain over unimodal training— our approach achieved substantial improvements. Finally, compared with unimodal learning, the proposed Pro-

---

[3]For consistency, we adopted some baseline results from Yang et al. (2024).

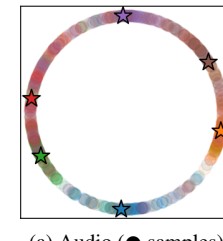 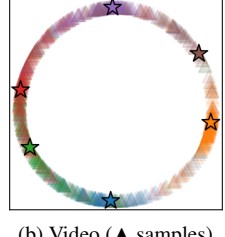 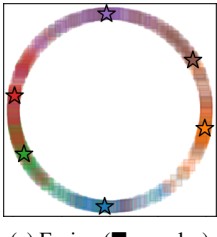

(a) Audio (● samples)  (b) Video (▲ samples)  (c) Fusion (■ samples)

Figure 4: **Visualization of unimodal and fused representations on a hypersphere** (CREMA-D test). ★ indicates a class prototype, and different colors indicate features from different classes.

MoBal improved accuracy by more than $4\%$ and up to $21\%$ across all datasets, underscoring the effectiveness of multimodal representation learning via ProMoBal.

### 4.3 VISUALIZATION OF PROMOBAL REPRESENTATIONS

Figure 4 visualizes unimodal and fused representations produced by the ProMoBal framework on the unit hypersphere. This visualization demonstrates that proposed ProMoBal can promote the new representational geometry illustrated in Figure 1(b). Specifically, *(1)* unimodal representations are encouraged to move toward their shared prototypes, while *(2)* the prototypes themselves are promoted to align in a regular simplex geometry. At the same time, *(3)* fused representations are encouraged to align with their target prototypes while preserving the relative geometry among all prototypes. We expect that such a configuration reduces modality imbalance by selecting prototype-consistent features at each dimension, thereby preventing dominant modalities from overwhelming the fused space. We also expect that in the fused embedding space, decision boundaries become smoother. We show in §B of the appendix that fused representations exhibit stronger feature collapse and a clearer regular simplex geometry than unimodal ones.

### 4.4 ANALYSIS OF MODALITY CONTRIBUTIONS IN PROMOBAL

Table 2 examines whether ProMoBal effectively balances modality contributions by quantifying how often each modality is selected by the proposed SaMoBal fusion module (summing to $100\%$). The selection rates

Table 2: Empirical analysis of modality contributions

| Datasets | CREMA-D | | NVGesture | | |
|---|---|---|---|---|---|
| Modalities | Audio | Video | RGB | OF | Depth |
| Selection rate (%) | 51.09 | 48.91 | 33.08 | 33.03 | 33.89 |

across modalities are well balanced—even in the three-modality setup—indicating that no single modality dominates or is suppressed. In other words, ProMoBal ensures fair use of all modalities, preventing imbalance and fully leveraging their complementary information in fusion.

## 5 CONCLUSION

The rapid proliferation of multimodal sensors in real-world applications demands frameworks that can effectively integrate diverse modalities. As their number and diversity grow, maintaining balanced and geometrically consistent representations becomes a critical challenge.

In this work, we proposed **ProMoBal**, an E2E MSCL framework that promotes our new *regular simplex geometry* tailored for multimodal representation learning, where *shared prototypes* serve as a common reference for both unimodal and fused embeddings. Together with its *sample-adaptive fusion* mechanism, ProMoBal promotes balanced contributions across modalities, preventing dominance by stronger modalities. Through its key components—Pro-UCL, SaMoBal, Pro-FD, and CA—ProMoBal encourages complementary outcomes: promoting modality and class balance, preserving modality-specific discrimination, capturing modality-invariant information, and encouraging geometry-consistent embeddings. Our extensive experiments with five benchmark datasets show that ProMoBal consistently outperforms SOTA multimodal supervised learning methods. We believe these findings highlight the potential of geometry-aware design and sample-adaptive fusion as a foundation for future multimodal learning research. For future work, we plan to investigate the computational costs of ProMoBal, which grow with the number of modalities and classes.

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

# ProMoBal: Prototype-guided Modality Balancing in multimodal contrastive learning (Appendix)

For clarify and completeness, this appendix provides additional details, analysis, and results:

- §A details our experimental setups, including datasets and implementations.
- §B quantitatively analyzes unimodal and fusion representations in the perspective of the proposed regular simplex geometry.
- §C reports ablation studies examining the contribution of each component.
- §D evaluates variants of the SaMoBal module with different selection schemes.
- §E compares saliency-based visual explanations from the proposed ProMoBal framework against other multimodal learning methods.

## A   DETAILS OF EXPERIMENTAL SETUPS

### A.1   DATASET DETAILS

This section provides details of the five benchmark datasets used in this study:

- The **CREMA-D** audio–video dataset (Cao et al., 2014), designed for emotion recognition with six categories, consists of 7,442 clips, including 6,698 for training and 744 for testing.
- The **KineticsSounds** audio–video dataset (Arandjelovic & Zisserman, 2017), designed for action recognition with 31 categories, contains 19,000 video clips, with 15,000 for training, 1,900 for validation, and 1,900 for testing.
- The **Sarcasm** image-text dataset (Cai et al., 2019), designed for sarcasm detection with two categories, includes 24,635 image–text pairs, divided into 19,816 for training, 2,410 for validation, and 2,409 for testing.
- The **Twitter2015** image-text dataset (Yu & Jiang, 2019), designed for emotion recognition with three categories, consists of 5,338 image–text pairs, with 3,179 for training, 1,122 for validation, and 1,037 for testing.
- The **NVGesture** RGB-depth-OF dataset (Molchanov et al., 2016), designed for dynamic hand gesture recognition with 25 categories, contains 1,532 samples, with 1,050 for training and 482 for testing.

### A.2   IMPLEMENTATION DETAILS

This section provides implementation details. We conducted all experiments using PyTorch v2.0.0 (Paszke et al., 2017) and NVIDIA GeForce RTX 4090 GPUs.

#### A.2.1   UNIMODAL ENCODERS

For all datasets, we followed the unimodal encoder architectures in Yang et al. (2024):

- For the audio–video datasets, CREMA-D and KineticsSounds, we used ResNet-18 (He et al., 2016) as the backbone and set the feature dimension $D$ to 512.
- For the image–text datasets, Sarcasm and Twitter2015, we employed ResNe-50 (He et al., 2016) for image encoding and BERT (Devlin et al., 2019) for text processing, with $D = 1024$.
- For the RGB–depth–OF dataset, NVGesture, we adopted I3D (Carreira & Zisserman, 2017) as the unimodal encoder and set $D = 1024$.

#### A.2.2   HYPERPARAMETERS FOR THE PROPOSED FRAMEWORK

**Optimization parameters.** We followed the following optimization setup in (Yang et al., 2024):

- For the audio–visual (i.e., CREMA-D and KineticsSounds) and RGB–depth–OF datasets (i.e., NVGesture), we used stochastic gradient descent with momentum 0.9 and weight decay 0.1.
- For the image–text datasets (i.e., Sarcasm and Twitter2015), we used the Adam optimizer with an initial learning rate of $2 \times 10^{-5}$.

Table A1: Accuracy comparisons of ProMoBal under varying balancing parameter combinations (in %).

| Balancing parameters | | | Datasets | | | | |
|---|---|---|---|---|---|---|---|
| $\mathcal{L}_{\text{uni}}$ | $\mathcal{L}_{\text{fus}}$ | $\mathcal{L}_{\text{CA}}$ | CREMA-D | KineticsSounds | Sarcasm | Twitter2015 | NVGesture |
| 1 | 1 | 1 | 84.93 | 76.53 | 88.75 | 76.57 | 85.06 |
| 0.7 | 1 | 1 | 83.96 | 75.92 | 88.71 | 76.92 | 84.95 |
| 1 | 3 | 1 | 84.26 | 75.24 | 88.59 | 76.66 | 86.27 |
| 0.7 | 3 | 1 | 85.27 | 74.86 | 88.38 | 76.18 | 85.94 |
| 1 | 1 | 3 | 84.70 | 75.24 | **89.17** | **77.08** | 86.02 |
| 0.7 | 1 | 3 | **85.68** | 75.27 | 88.46 | 75.85 | **86.84** |
| 1 | 3 | 3 | 84.53 | **76.66** | 88.42 | 76.35 | 85.65 |
| 0.7 | 3 | 3 | 84.12 | 76.11 | 88.04 | 75.51 | 85.15 |

- For all datasets, the learning rate was decayed by a factor of $0.1$ at epoch 70.

We set the initial learning rate to $0.01$ for CREMA-D and NVGesture, and $0.1$ for KineticsSounds. We set the batch size $B$ as follows: for CREMA-D, $B = 16$; for KineticsSounds, Sarcasm, and Twitter2015, $B = 64$; and for NVGesture, $B = 4$. We trained the ProMoBal model for $150$ epochs, for all datasets.

**Balancing parameters among proposed losses.** In tuning the balancing parameters $\{\lambda_{\text{uni}}, \lambda_{\text{fus}}, \lambda_{\text{CA}}\}$ in §3.5, we evaluated the corresponding training losses $\{\lambda_{\text{uni}}\mathcal{L}_{\text{uni}}, \lambda_{\text{fus}}\mathcal{L}_{\text{fus}}, \lambda_{\text{CA}}\mathcal{L}_{\text{CA}}\}$ at the initial epoch and selected parameter values that ensured balanced contributions, i.e., $\lambda_{\text{uni}}\mathcal{L}_{\text{uni}} \approx \lambda_{\text{fus}}\mathcal{L}_{\text{fus}} \approx \lambda_{\text{CA}}\mathcal{L}_{\text{CA}}$. We fixed $\lambda_{\text{CE}} = 1$ throughout all experiments. The parameter combinations selected through this scheme are as follows:

- For the CREMA-D and NVGesture datasets, $\{\lambda_{\text{uni}} = 0.7, \lambda_{\text{fus}} = 1, \lambda_{\text{CA}} = 3\}$;
- For the Sarcasm and Twitter2015 datasets, $\{\lambda_{\text{uni}} = 1, \lambda_{\text{fus}} = 1, \lambda_{\text{CA}} = 3\}$;
- For the KineticsSounds dataset, $\{\lambda_{\text{uni}} = 1, \lambda_{\text{fus}} = 3, \lambda_{\text{CA}} = 3\}$;

Using the aforementioned tuning scheme, we consistently observed reliable classification performance across all benchmark datasets. Table A1 reports inference accuracy (in %) with different balancing parameter combinations. ***Note*** *that across all balancing parameter combinations in Table A1, the proposed ProMoBal consistently outperformed existing multimodal learning baselines.*

### A.2.3 HYPERPARAMETERS FOR EXISTING MULTIMODAL LEARNING BASELINES

For fair comparisons, we followed the default settings of existing multimodal learning baselines, except for the initial learning rate for GMC and DI-MML. To achieve the best performance for each dataset, we tuned the learning rates of GMC and DI-MML. Specifically, we selected initial learning rates of $\{10^{-1}, 10^{-1}, 2 \times 10^{-5}, 2 \times 10^{-5}, 10^{-2}\}$ for GMC and $\{10^{-3}, 10^{-1}, 2 \times 10^{-5}, 2 \times 10^{-5}, \text{N/A}\}$ for DI-MML, corresponding to the CREMA-D, KineticsSounds, Sarcasm, Twitter2015, and NVGesture datasets, respectively. Note that we could not apply DI-MML to the NVGesture dataset, as it does not support three modalities.

### A.3 DATA PREPARATION

For fair comparisons, we followed the preprocessing procedures used in prior works:

- For the audio–video datasets, CREMA-D and KineticsSounds, we adopted the preprocessing schemes of Yang et al. (2024) and Fan et al. (2024). We converted audio data into spectrograms of size $257 \times 299$ for CREMA-D and $257 \times 1004$ for KineticsSounds using the `librosa` library (McFee et al., 2015).
- For the image–text datasets, Sarcasm and Twitter2015, we followed the preprocessing procedures in Yang et al. (2024). We resized all images to $224 \times 224$ and truncated text sequences to a maximum length of $128$ tokens.
- For the NVGesture dataset, we adopted the preprocessing protocols of Wu et al. (2022) and Yang et al. (2024). We randomly sampled $64$ consecutive frames from each video, zero–padding sequences shorter than $64$ frames, and resized all frames to $224 \times 224$.

Table A2: Analysis of fused and unimodal representations (the ↓ and ↑ symbols indicate that the lower the better and higher the better, respectively).

| Metrics | Datasets | | | | | | |
| --- | --- | --- | --- | --- | --- | --- | --- |
| | CREMA-D | | | NVGesture | | | |
| | Audio | Video | Fusion | RGB | OF | Depth | Fusion |
| FC in (A1)$^{\downarrow}$ | 0.55 | 0.62 | **0.51** | 0.60 | 0.63 | 0.59 | **0.56** |
| RS in (A2)$^{\uparrow}$ | 0.81 | 0.78 | **0.83** | 0.81 | 0.78 | 0.85 | **0.88** |

Table A3: Ablation study for the primary components of the proposed ProMoBal (ACC (%)).

| | Components | | | | Datasets | | | | | | |
| --- | --- | --- | --- | --- | --- | --- | --- | --- | --- | --- | --- |
| | $\mathcal{L}_{\text{uni}}$ | SaMoBal | $\mathcal{L}_{\text{fus}}$ | $\mathcal{L}_{\text{CA}}$ | CREMA-D | | | NVGesture | | | |
| | | | | | Audio | Video | Fusion | RGB | OF | Depth | Fusion |
| No unimodal representation learning | × | ○ | ○ | ○ | 65.13 | 78.54 | 82.53 | 76.76 | 77.39 | 83.22 | 85.48 |
| No fused representation learning | ○ | × | ○ | ○ | 65.73 | 72.61 | 71.27 | 77.90 | 73.55 | 82.97 | 83.76 |
| | ○ | ○ | × | ○ | 66.53 | 78.96 | 83.42 | 77.88 | 74.07 | 83.82 | 86.39 |
| | ○ | × | × | ○ | 64.38 | 77.82 | 68.41 | 78.01 | 74.92 | 83.46 | 82.37 |
| No classifier alignments | ○ | ○ | ○ | × | 67.24 | 72.32 | 73.70 | 76.65 | 74.61 | 82.71 | 83.02 |
| **ProMoBal** | ○ | ○ | ○ | ○ | **68.07** | **80.91** | **85.68** | **78.62** | **79.01** | **84.16** | **86.84** |

# B  UNIMODAL VS. FUSED REPRESENTATION ANALYSIS

This section analyzes fused representations in comparison with unimodal ones. We employed two widely used metrics, alignment and uniformity (Chen et al., 2020; He et al., 2020; Li et al., 2022; Wang & Isola, 2020). In this study, we denote alignment as the Feature Collapse (FC) metric and uniformity as the Regular Simplex (RS) metric.

**Feature Collapse:** The FC metric averages distances between features from the same class (Wang & Isola, 2020; Li et al., 2022):

$$\text{FC} := \frac{1}{C} \sum_{c=1}^{C} \frac{1}{|\mathcal{F}_c|^2} \sum_{\mathbf{z}^i, \mathbf{z}^j \in \mathcal{F}_c} \left\| \mathbf{z}^i - \mathbf{z}^j \right\|_2, \tag{A1}$$

where $\mathcal{F}_c$ denotes the set of all features belonging to $c$th class. This metric measures how the features from the same class align to the class means. A smaller FC value indicates greater compactness among samples within the same class.

**Regular Simplex:** The RS metric averages distances between different class means in the representation space (Wang & Isola, 2020; Li et al., 2022):

$$\text{RS} := \frac{1}{C(C-1)} \sum_{c=1}^{C} \sum_{\substack{c'=1 \\ c' \neq c}}^{C} \left\| \boldsymbol{\mu}_c - \boldsymbol{\mu}_{c'} \right\|_2, \tag{A2}$$

where $\boldsymbol{\mu}_c$ denotes the $c$th class mean. This metric evaluates how well the class means are uniformly distributed on a unit hypersphere, i.e., the degree of a regular simplex configuration. The higher RS value, the better separation between different classes, i.e., closer to a regular simplex configuration.

The results in Table A2 show that fused representations exhibit stronger feature collapse and a clearer regular simplex geometry than unimodal ones, supporting our argument in §4.3. This implies that fusion effectively leverages complementary modality information to form more compact and uniformly separated class representations.

# C  ABLATION STUDIES

Ablation studies in Table A3 examine the contribution of each component in ProMoBal by analyzing classification accuracy of unimodal and fused embeddings. To quantify their impact, we

Table A4: Comparisons between different selection schemes in the proposed SaMoBal module.

| Selection schemes | Datasets | | | |
|---|---|---|---|---|
| | CREMA-D | | NVGesture | |
| | ACC (%) | MAP (%) | ACC (%) | $F_1$ (%) |
| Hard | 84.57 | 91.11 | 85.27 | 85.42 |
| Soft | 82.98 | 90.06 | 83.85 | 84.22 |
| **Ours** | **85.68** | **91.85** | **86.84** | **86.97** |

systematically removed each component and grouped the analysis into three categories: unimodal representation learning, fused representation learning, and classifier alignments.

**Unimodal representation learning:** Removing proposed $\mathcal{L}_{uni}$ in §3.2.1 substantially degrades unimodal accuracy and weakens fusion performance, as unimodal features no longer achieve a common prototype-based representation goal.

**Fused representation learning:** We replaced the proposed SaMoBal module $\Phi$ in §3.3.1 and §3.3.3 with simple concatenation and substituted proposed $\mathcal{L}_{fus}$ in §3.3 with the standard CE loss. Without SaMoBal, weaker modalities are suppressed due to the absence of sample-level balancing. Without $\mathcal{L}_{fus}$, the fused embeddings are only aligned to the GT prototype but fail to capture the relational geometry among all prototypes, leading to degraded fusion accuracy.

**Classifier alignments:** Removing proposed $\mathcal{L}_{CA}$ in §3.4 lowers overall performance, as the model does not strongly induces a symmetric geometric configuration between the representation and classifier spaces.

Overall, Table A3 shows that the best performance arises when all components are integrated together, underscoring the synergistic effect of the proposed framework.

# D ANALYSIS OF DIFFERENT SELECTION SCHEMES IN THE SAMOBAL MODULE

This section analyzes variants of the SaMoBal module, particularly using different selection schemes on a *per-sample* basis: "hard", "soft", and ours in Figure 3. We describe each selection scheme with the training perspective as follows (for simplicity):

- In the **hard selection** scheme, we constructed fused embeddings by choosing the modality feature with the highest similarity to its corresponding prototype in each dimension. We replaced the Gumbel-Softmax and straight-through estimation (STE) trick in SaMoBal with a standard softmax.
- In the **soft selection** scheme, we computed fusion embeddings by weighted averaging, combining unimodal features in proportion to their cosine similarity scores. Specifically, for the $d$th dimension, given similarity scores $\{s_{1,d}, \ldots, s_{M,d}\}$ with the prototype, we compute the weights as $(w_{1,d}, \ldots w_{M,d}) = \mathrm{Softmax}(s_{1,d}, \ldots, s_{M,d})$, and then obtain the fusion feature as $(\mathbf{z}_{fus})_d = \sum_{m=1}^{M} w_{m,d}(\mathbf{z}_m)_d, d = 1, \ldots, D$. We apply this procedure at the sample level.
- **Ours** corresponds to the proposed selection scheme in §3.3.1.

In inference, we used the proposed inference scheme (3) consistently to the hard and soft selection schemes.

From the results in Table A4, we draw two key observations:

- **Hard selection is more effective than soft selection:** Hard selection proved more effective than soft selection by focusing on the modality feature most consistent with the prototype at each dimension. This suggests that hard selection yields fusion embeddings that are more discriminative while maintaining modality invariance.
- **Approximating hard selection in a differentiable manner is effective:** The approximation of hard selector using the Gumbel-softmax and STE trick (see §3.3.1) allows the model to retain the benefits of discrete, prototype-consistent feature selection, while enabling end-to-end optimization through gradient backpropagation. This combination yields more discriminative and

modality-invariant fusion embeddings, ultimately improving multimodal classification performance.

## E SALIENCY-BASED VISUAL EXPLANATIONS BETWEEN DIFFERENT MULTIMODAL SUPERVISED LEARNING METHODS

We used Grad-CAM (Selvaraju et al., 2017) to visualize and explain the image regions attended by the visual modality. Grad-CAM assigns importance scores to each pixel in a feature map, thereby providing visual explanations of the regions most critical for the model's predictions.

Figure A1 compares visual explanations obtained with simple concatenation, DI-MML (Fan et al., 2024), and the proposed ProMoBal, with the captions under each image indicating the corresponding paired text. The proposed ProMoBal framework attends more precisely to image regions aligned with the semantic cues in the paired text. This demonstrates that our method effectively exploits both text and visual modalities, guiding them toward a common goal and thereby enabling more effective multimodal learning.

| Image | Concat. | DI-MML | **ProMoBal (ours)** |

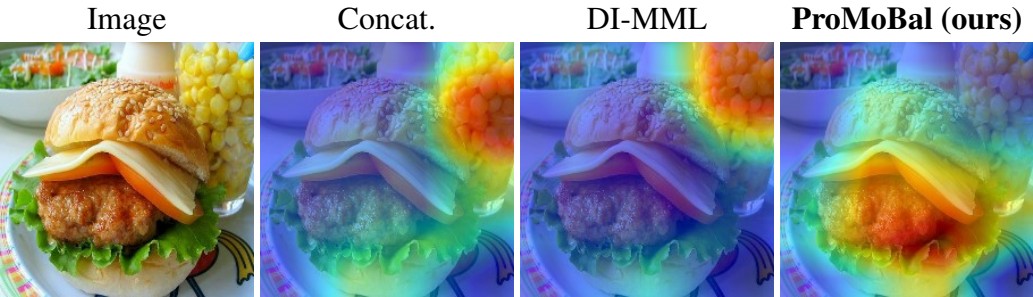

(a) The most delicious breakfast in the world ! So love sandwich ! ! Lint . . . . $T$ also liked it !

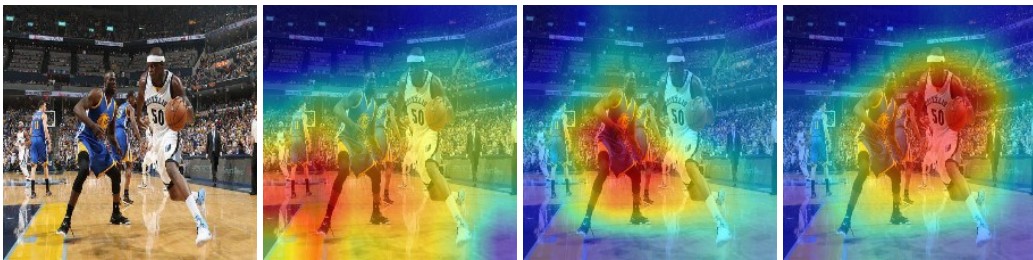

(b) RT @ SLAMonline : $T$ Drops 22 Points in Game 3 Win Over Warriors

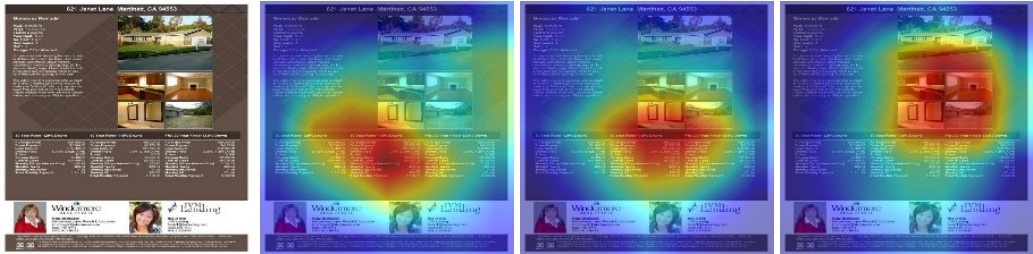

(c) Great Opportunity in $T$ , CA ! ! ! Check it out ! Buenos Dias ! ! !

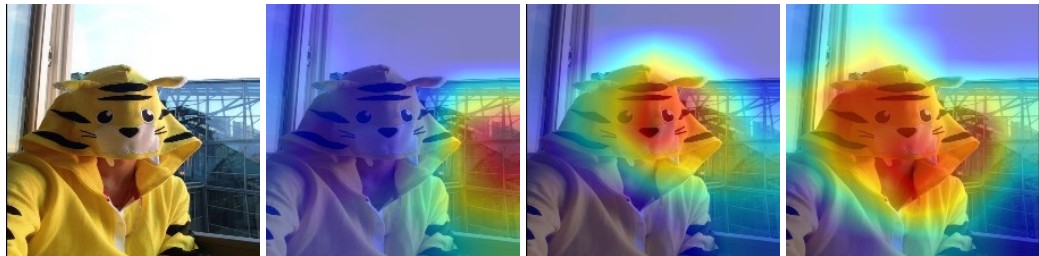

(d) RT @ MileyCyrus : Tigger in $T$ Germany

Figure A1: Comparisons of saliency-based visual explanations between different multimodal supervised learning methods (Twitter2015 dataset). The captions indicate the text cues paired with each image.

