# OpenReview forum: "ProMoBal: Prototype-guided Modality Balancing in multimodal contrastive learning"
_ICLR.cc/2026/Conference — ICLR 2026 Conference Withdrawn Submission_

### Official Review · Reviewer_g5PT · 2025-10-19

**Soundness:** 3
**Presentation:** 2
**Contribution:** 2
**Rating:** 2
**Confidence:** 4

**Summary:**

The paper addresses modality imbalance in multimodal learning (dominant modalities overshadow weaker ones) and proposes ProMoBal, an end-to-end MSCL framework. ProMoBal integrates prototype-centered learning and sample-adaptive fusion, promoting a regular simplex geometry with class prototypes symmetrically on a shared hypersphere. Its core components include Pro-UCL, SaMoBal, Pro-FD, and CA， aiming at aligning unimodal and fused embeddings while preserve modality-specific and modality invariant features. Experiments on five datasets show ProMoBal outperforms SOTA methods.

**Strengths:**

1. This paper introduces the regular simplex geometry centered on shared class prototypes， which is a novel configuration that unifies unimodal and fused embeddings under a common reference, resolving the longstanding objective mismatch between unimodal and fused representation learning.
2. SaMoBal module creatively addresses per-sample modality variability by adjusting the fusion at both the sample and dimension levels, a departure from conventional uniform fusion strategies.
3. This paper includes rigorous design choices, using Gumbel-Softmax with straight-through estimation to make SaMoBal’s hard selection differentiable
4. Empirically, this paper supports claims with comprehensive experiments on five datasets spanning audio-video, image-text, and three-modality tasks.

**Weaknesses:**

1. The mismatch between unimodal and fused representation learning should be discussed in detail with theoretical or experimental demonstration.
2. Shared prototypes seem to be the key to the design of this method. However, explanations for the rationale behind selecting such prototypes, how to calculate these prototypes, and the feasibility of their application are lacking.
3. Why is it claimed that SoMoBAL can balance the learning of fused representations? If Modality 2 dominates, can it be considered that Modality 2 is closer to the shared prototypes? Then, will taking the prototypes as the learning target not lose the specific information of Modality 1?
4. There is a relatively large number of hyperparameters, and no experimental analysis has been conducted on their sensitivity.
5. This paper introduces a substantial amount of additional computation, such as the calculation of $z_{fus}$ and more loss functions, yet it lacks an analysis of computational complexity and experimental comparisons regarding the algorithm’s computational cost.
6. This paper claims to achieve a better class balance effect and can be experimentally verified on datasets with class imbalance.
7. The structure and writing of this paper need improvement. Excessively similar nouns (such as "shared class prototype" and "class prototype") as well as an overabundance of symbols and bold text have hindered the reading fluency.

**Questions:**

None

---

### Official Review · Reviewer_17K3 · 2025-10-29

**Soundness:** 3
**Presentation:** 2
**Contribution:** 2
**Rating:** 4
**Confidence:** 3

**Summary:**

This paper focuses on the problem of modality imbalance in multimodal learning, where dominant modalities often overshadow weaker ones during feature fusion, and unimodal encoders may lack a common representational objective. To tackle these issues, the paper introduces ProMoBal (Prototype-guided Modality contribution Balancing), which is an end-to-end multimodal supervised contrastive learning (MSCL) framework. The basic idea is to establish a shared regular simplex geometry centered around class prototypes, which serves as a common goal for both unimodal and fused representations. The proposed ProMoBal is evaluated on five benchmark classification datasets spanning audio-video, image-text, and RGB-depth-optical flow modalities. The results demonstrate consistent state-of-the-art performance, outperforming various multimodal supervised learning methods, including other MSCL approaches.

**Strengths:**

- Clear motivation: the paper aims to solve the issues of objective mismatch and sample-level variation in modality importance;
- The framework achieves state-of-the-art results across five diverse benchmark datasets, demonstrating its effectiveness;

**Weaknesses:**

- The inference procedure for SaMoBal seems computationally expensive and potentially complex. It requires generating C candidate fusion embeddings (one for each possible class prototype) and then selecting the best one based on alignment scores. How does this scale with a large number of classes (C)?
- The proposed approach relies heavily on shared class prototypes ($P$). However, it doesn't explicitly state how these prototypes are initialized or learned. Are they learnable parameters updated via backpropagation, fixed templates, or perhaps moving averages of features?
- The overall loss function needs four balancing hyperparameters ($\lambda_{uni}, \lambda_{CE}, \lambda_{fus}, \lambda_{CA}$. Does the framework's performance might be sensitive to these choices?
- Minor typos & suggestions: L20-L21: “mitigates modality balance” should be “mitigates modality imbalance”; L691: “ResNe-50” should be “ResNet-50”;
- One open question & suggestion: The experiments are mainly conducted on classification tasks using relatively standard benchmark datasets. While demonstrating SOTA is valuable, exploring ProMoBal's effectiveness on other types of multimodal tasks (e.g., regression, retrieval, generation) is also helpful.

Overall, the main concern is about the computational complexity. I would like to see more author rebuttal in terms of this part.

**Questions:**

Please see weakness & questions.

---

### Official Review · Reviewer_2HGH · 2025-10-31

**Soundness:** 2
**Presentation:** 2
**Contribution:** 3
**Rating:** 4
**Confidence:** 5

**Summary:**

The paper introduces a multimodal supervised contrastive learning method, called ProMoBal, which addresses the problem of modality imbalance and contributions of specific unimodal samples. The method consists of 4 components; Prototype-based Unimodal Contrastive Learning (Pro-UCL) loss, the Sample-Adaptive Modality Contribution Balancing (SaMoBal) fusion module, Prototype–Fusion Distillation (Pro-FD) loss, and Classifier Alignment (CA) loss for each modality classifier. Experiments are performed on five benchmark datasets.

**Strengths:**

Interesting idea to align unimodal and fused representations via prototypes

**Weaknesses:**

- The method is complicated
- Hard to extract the relevant details from the main paper
- Reliance on the class label during training and necessity to have all modalities during inference
- Seems hard to tune the many loss terms and their parameters, unclear how easy it is to obtain the prototypes

**Questions:**

- it is not clear to me what is the difference between $p_c$ and $p_y$  and $z_{fus}$ in section 3.3.2. How do you even obtain these prototypes? What do you take as ground truth prototypes, how are these initialized? Similarly ,there are no details about the prototype bank $P$ in section 3.3.3. What is this, how big is it?
- The proposed method seems to require access to all modalities during inference, which is contrary to the compared methods like GMC, DI-MML, MLA etc. This seems like a major shortcoming of the method as partial inference isn’t possible - or at least there is no experiment like this in the paper. Which experiment in the paper really demonstrates the added value of ProMoBal? As a baseline, one could as well just infer the class from the modality-specific classifiers or use the ProMoBal without the fusion part. In general, there are limited ablation studies on all the different loss terms and how to tune these parameters.
- How does the runtime of ProMoBal compare to other methods?
- The method requires a labelled dataset for training. Do you assume uniform distribution over classes? What happens when there are class imbalances, especially with the prototypes (which are not explained)?

Despite promising results in Table 1, I decided for a lower score based on the following:
- No information regarding the class prototypes and the prototype bank.
- No experiments performed on partial/missing modality setups and compared with existing methods.

---

### Official Review · Reviewer_BzF5 · 2025-11-01

**Soundness:** 2
**Presentation:** 2
**Contribution:** 2
**Rating:** 2
**Confidence:** 5

**Summary:**

The paper proposes an end-to-end multimodal supervised contrastive learning framework named ProMoBal (Prototype-guided Modality Balancing). This framework centers on "shared class prototypes" as a core reference, constructing a "regular simplex geometric structure" (where class prototypes are symmetrically and equidistantly distributed on a shared hypersphere). It achieves balanced representation learning through three key components: 1) The Pro-UCL loss, which guides unimodal embeddings to cluster around shared prototypes while preserving modality-specific characteristics; 2) The SaMoBal module, which integrates Gumbel-Softmax to enable adaptive modality selection at both sample and feature levels, preventing dominant modalities from overshadowing weaker ones; 3) The Pro-FD loss and CA loss, which distill inter-prototype geometric relationships into fused embeddings and align unimodal classifiers with the prototypes, respectively.

**Strengths:**

1. The motivation is clear and convincing.

2. The framework achieves performance improvements across five datasets of varying modality types, including audio-visual pairs (CREMA-D, KineticsSounds), image-text pairs (Sarcasm, Twitter2015), and the trimodal NVGesture. Notably, it excels even in the trimodal setting, underscoring its generalizability regardless of modality quantity or combination.

**Weaknesses:**

1. While the paper initially aims to address the conflict between unimodal and multimodal objectives, the proposed Prototype-based Unimodal Contrastive Learning does not appear to resolve this issue. Specifically, the Cross-Entropy (CE) loss encourages each modality to learn distinct, discriminative features, whereas the Prototype-based loss pushes them toward a common, shared representation. This creates a fundamental tension between the two learning signals, leaving the core conflict between specialization and sharing unaddressed.

2. The experimental settings appear to be inconsistent and unfair. For instance, the learning rate and number of training epochs differ from those used in existing methods, such as MMPareto, where a learning rate of 1e-3 and 100 training epochs were adopted. The authors should reproduce the comparisons under the same settings as ProMoBal. Additionally, the reported unimodal performance (CREMA-D and KineticsSounds) also seems to be based on previous configurations rather than those specified in this paper. In fact, if the learning rate is 1e-2 and the training epoch is 150, the unimodal performance of CREMA-D can be higher than 0.7. These issues raise concerns about the credibility of the experimental results presented.

3. The description of the proposed method is not sufficiently clear. For instance, the process for obtaining the prototypes is not explained. Furthermore, the current notation could be simplified to enhance readability.

4. The authors should show the unimodal performance in the results, which can help verify the effectiveness of ProMoBal to improve unimodal learning in multimodal learning.

5. The authors should compare ProMoBal with D&R[1], which also considers the effect of modality quality.





[1] Diagnosing and Re-learning for Balanced Multimodal Learning ECCV2024

**Questions:**

See the weakness

---

### Note · Authors · 2025-11-27

I have read and agree with the venue's withdrawal policy on behalf of myself and my co-authors.